# Generalization Bounds for Gradient Methods via Discrete and Continuous Prior

**Xuanyuan Luo**
IIIS, Tsinghua University
xuanyuanluo@google.com

**Luo Bei**
Renmin University of China
rabbit_lb@ruc.edu.cn

**Jian Li**
IIIS, Tsinghua University
lijian83@mail.tsinghua.edu

## Abstract

Proving algorithm-dependent generalization error bounds for gradient-type optimization methods has attracted significant attention recently in learning theory. However, most existing trajectory-based analyses require either restrictive assumptions on the learning rate (e.g., fast decreasing learning rate), or continuous injected noise (such as the Gaussian noise in Langevin dynamics). In this paper, we introduce a new discrete data-dependent prior to the PAC-Bayesian framework, and prove high probability generalization bounds of order $O(\frac{1}{n} \cdot \sum_{t=1}^{T} (\gamma_t/\varepsilon_t)^2 \|\mathbf{g}_t\|^2)$ for floored GD and SGD (i.e. finite precision versions of GD and SGD with precision level $\varepsilon_t$) where, where $n$ is the number of training samples, $\gamma_t$ is the learning rate at step $t$, $\mathbf{g}_t$ is roughly the difference between the average gradient over all samples and that over only prior samples. $\|\mathbf{g}_t\|$ is upper bounded by (typically much smaller) than the gradient norm $\|\nabla f(W_t)\|$. We remark that our bounds hold for nonconvex and nonsmooth loss functions. Moreover, our theoretical results provide numerically favorable upper bounds of testing errors (0.026 on MNIST and 0.198 on CIFAR10). Furthermore, we study the generalization bounds for gradient Langevin Dynamics (GLD). Using the same framework with a carefully constructed continuous prior, we show a new high probability generalization bound of order $O(\frac{1}{n} + \frac{L^2}{n^2} \sum_{t=1}^{T} (\gamma_t/\sigma_t)^2)$ for GLD. The new $1/n^2$ rate is obtained using the concentration of the difference between the gradient of training samples and that of the prior.

## 1 Introduction

Bounding generalization error of learning algorithms is one of the most important problems in machine learning theory. Formally, for a supervised learning problem, the generalization error is defined as the testing error (or population error) minus the training error (or empirical error). In particular, we denote $\mathcal{R}(w, (x, y)) := \mathbb{1}[h_w(x) \neq y]$ as the error of a single data point $(x, y)$, where $h_w(x)$ is the output of a model with parameter $w \in \mathbb{R}^d$. Suppose $S$ is the set of training data, each i.i.d. sampled from the population distribution $\mathcal{D}$, and we use $\mathcal{R}(w, S) := \frac{1}{|S|} \sum_{z \in S} \mathcal{R}(w, z)$ and $\mathcal{R}(w, \mathcal{D}) := \mathbb{E}_{z \sim D}[\mathcal{R}(w, z)]$ to denote the training error and the testing error, respectively. The generalization error of $w$ is formally defined as $\mathrm{err}_{\mathrm{gen}}(w) = \mathcal{R}(w, \mathcal{D}) - \mathcal{R}(w, S)$.

Proving tighter generalization bounds for general nonconvex learning and particularly deep learning has attracted significant attention recently. While the classical learning theory (uniform convergence theory) which bounds the generalization error by various complexity measures (e.g., the

36th Conference on Neural Information Processing Systems (NeurIPS 2022).

VC-dimension and Rademacher complexity) of the hypothesis class has been successful in several classical convex learning models, however, they become vacuous and hence fail to explain the success of modern nonconvex over-parametrized neural networks (i.e., the number of parameters significantly exceeds the number of training data) (see e.g., Zhang et al. [2017], Nagarajan and Kolter [2019]). Recently, learning theorists have tried to understand and explain generalization of deep learning from several other perspectives, such as margin theory [Bartlett et al., 2017, Wei et al., 2019], algorithmic stability [Hardt et al., 2016, Mou et al., 2018, Li et al., 2020, Bousquet et al., 2020], PAC-bayeisan [London, 2017, Bartlett et al., 2017, Neyshabur et al., 2018, Zhou et al., 2019, Yang et al., 2019], neural tangent kernel [Jacot et al., 2018, Du et al., 2019, Arora et al., 2019, Cao and Gu, 2019], information theory [Pensia et al., 2018, Negrea et al., 2019], model compression [Arora et al., 2018, Zhou et al., 2019], differential privacy [Oneto et al., 2017, Wu et al., 2021] and so on.

In this paper, we aim to obtain tighter generalization error bounds that depend on both the training data and the optimization algorithms (a.k.a. gradient-type methods) for general nonconvex learning problems. In particular, we prove algorithm-dependent generalization bounds for several gradient-based optimization algorithms such as certain variants of gradient descent (GD), stochastic gradient descent (SGD) and stochastic gradient Langevin dynamics (SGLD). Our proofs are based on the classic Catoni's PAC-Bayesian framework [Catoni, 2007] and also have a flavor of algorithmic stability [Bousquet and Elisseeff, 2002]. Several prior works have obtained generalization bounds for SGD and SGLD by analyzing trajectory through either the PAC-Bayesian or the algorithmic stability framework (or closely related information theoretic arguments). However, most existing results based on analyzing the optimization trajectories require either restrictive assumptions on the learning rates, or continuous noise (such as the Gaussian noise in Langevin dynamics) in order to bound the stability or the KL-divergence. In this paper, we resolve the above restrictions by combining the PAC-Bayesian framework with a few simple (yet effective) ideas, so that we can obtain new high probability and non-vacuous generalization bounds for several gradient-based optimization methods with either discrete or continuous noises (in particular certain variants of GD and SGD, either being deterministic or with discrete noise, which cannot be handled by existing techniques).

## 1.1 Prior work

We first briefly mention some recent work on bounding the generalization error of gradient-based methods. Hardt et al. [2016] first studied the uniform stability (hence the generalization) of stochastic gradient descent (SGD) for both convex and non-convex functions. Their results for non-convex functions requires that the learning rate $\eta_t$ scales with $1/t$. Their work motivates a long line of subsequent work on generalization error bounds of gradient-based optimization methods: Kuzborskij and Lampert [2018], London [2016], Chaudhari et al. [2019], Raginsky et al. [2017], Mou et al. [2018], Chen et al. [2018], Li et al. [2020], Negrea et al. [2019], Wang et al. [2021].

Recently, Simsekli et al. [2020], Hodgkinson et al. [2022] obtained generalization bound of SGD through the perspective of heavy-tailed behaviors and using the notion of Hausdorff dimension $d_{\mathrm{H}}$ which depends on both the algorithm and data.

**PAC-Bayesian bounds.** The PAC-Bayesian framework [McAllester, 1999] is a powerful method for proving high probability generalization bound [Bartlett et al., 2017, Zhou et al., 2019, Mou et al., 2018]. Roughly speaking, it bounds the generalization error by the KL divergence $\mathrm{KL}\left(Q \,\middle\|\, P\right)$, where $Q$ is the distribution of the learned output and $P$ is a prior distribution which is typically independent of dataset $S$. In this framework, bounding $\mathrm{KL}\left(Q \,\middle\|\, P\right)$ is the most crucial part for obtaining tighter PAC-Bayesian bounds. In order to bound the KL divergence, both the prior $P$ and posterior $Q$ are typically chosen to be continuous distributions (mostly Gaussians so that KL can be computed in closed form). Hence, most prior work either considered gradient methods with continuous noise (such as Gradient Langevin Dynamics) (e.g., [Mou et al., 2018, Li et al., 2020, Negrea et al., 2019]), or injected a Gaussian noise to the final parameter at the end (e.g., [Neyshabur et al., 2018, Zhou et al., 2019]) (so $Q$ is a Gaussian distribution). We also note that designing effective prior $P$ can be also very important. For example, Lever et al. [2013] proposed to use the population distribution to compute the prior. In fact, the prior can even partially depend on the training data [Parrado-Hernández et al., 2012, Negrea et al., 2019], and our Theorem 4.1 is partially inspired by this idea.

## 1.2 Our contributions

First, we provide high probability generalization bounds for *discrete* gradient methods. In particular, we study the generalization of Floored Gradient Descent (FGD), which is a variant of GD, and Floored Stochastic Gradient Descent (FSGD), a variant of SGD. We obtain our bound by an interesting construction of discrete priors. Secondly, we consider well studied gradient methods with continuous noise, (stochastic) gradient Langvin dynamics (GLD and SGLD). We show sharper generalization bounds by carefully bounding the concentration of the sample gradients. Now, we summarize our results.

**FGD and FSGD.** We first study an interesting variant of GD, called Floored GD (FGD) (Algorithm 1). The update rule of FGD is defined as follows:

$$W_t \leftarrow W_{t-1} - \gamma_t \nabla f(W_{t-1}, S_J) - \varepsilon_t \text{floor}\left(\gamma_t \mathbf{g}_t / \varepsilon_t\right), \tag{FGD}$$

where $S_J$ is the subset of training dataset $S$ with size $m$ indexed by subset $J \subset [n]$ ($J$ is chosen before training), $\nabla f(W_{t-1}, Z) := \frac{1}{|Z|} \sum_{z \in Z} \nabla f(W_{t-1}, z)$ is the average gradient over the dataset $Z$, $\gamma_t$ is the learning rate, $\varepsilon_t$ is the precision level, and $\mathbf{g}_t := \nabla f(W_{t-1}, S) - \nabla f(W_{t-1}, S_J)$ is the gradient difference. The flooring operation is defined by $\text{floor}(x) := \text{sign}(x)\lfloor|x|\rfloor$ for any real number $x$. FGD can viewed as GD with given precision limit $\varepsilon_t$. We can see if we ignore the floor operation or let $\varepsilon_t$ approaches 0, FGD reduces to GD (see also Appendix A).

We also study a finite precision variant of SGD, called Floored SGD (FSGD) (see Section 5 for its formal definition). Empirically, the optimization and generalization capabilities of FGD and FSGD are very close to those of GD and SGD (see Figure 5 and 6 in Appendix H).

By constructing a discrete data-dependent prior and incorporate it into Catoni's PAC-Bayesian framework, we prove that the following bound (Theorem 5.2) holds for FGD with high probability:

$$\mathcal{R}(W_T, \mathcal{D}) \leq c_0 \mathcal{R}(W_T, S_{[n]\setminus J}) + O\left(\frac{1}{n-m} + \frac{\ln(dT)}{n-m} \sum_{t=1}^{T} \frac{\gamma_t^2}{\varepsilon_t^2} \|\mathbf{g}_t\|^2\right),$$

where $d$ is the dimension of parameter space and $c_0$ can be chosen to be a small constant. The bound for FSGD is very similar (see Theorem 5.3). Now we make a few remarks about our results.

1. Our result holds for nonconvex and nonsmooth learning problems (replacing the gradients with subgradients for nonsmooth cases). Moreover, there is no additional requirement on the learning rate $\gamma_t$.

2. The gradient difference $\mathbf{g}_t$ is typical much smaller than the worst case gradient norm. It usually decreases when $m = |J|$ grows (see Figure 1c in Section 7).

3. We obtain non-vacuous generalization bounds on commonly used datasets. Specifically, our theoretical test error upper bounds on MNIST and CIFAR10 are **0.026** and **0.198**, respectively (see Section 7). Both of them are tighter than the best-known MNIST bound (11%) and CIFAR10 bound (23%) reported in Dziugaite et al. [2021]. See Table 1 in Appendix B for more comparisons.

4. In order to bound the KL between $P$ and the deterministic process of FGD, we construct the prior $P$ from a discrete random processes.. We hope it may inspire future research on handling deterministic optimization algorithms or discrete noise.

**Why study FGD/FSGD?** We would like to remark that we study FGD/FSGD, not because FGD/FSGD have better performances than GD/SGD or other advantages. Indeed, their performances are almost the same as those of GD/SGD (see Appendix H). We use them as important stepping stones to study generalization bounds for GD and SGD. Note that most existing trajectory-based generalization bounds require either fast decreasing learning rate, or continuous injected noise, such as the Gaussian noise in Langevin dynamics, for general non-convex loss functions. Handling deterministic algorithms (such as GD) or discrete noises (such as SGD) is challenging and beyond the reach of existing techniques. In fact, understanding such discrete noises and their effects on generalization has been an important research topic (see e.g., Li et al. [2020], Zhu et al. [2019], Ziyin et al. [2021]). In particular, Zhu et al. [2019] show that it is insufficient to approximate SGD's discrete noise by isotropic Gaussian noise. Moreover, proving nontrivial generalization bounds for SGD-like algorithms with discrete noise has also been proposed as an open research direction in Li et al. [2020].

**GLD and SGLD.** We provide a new generalization bound for Gradient Langevin Dynamics (GLD). The update rule of GLD is defined as follows.

$$W_t \leftarrow W_{t-1} + \gamma_t \nabla f(W_{t-1}, S) + \sigma_t \mathcal{N}(0, I_d). \tag{GLD}$$

In this paper, we show that the following generalization bound (Theorem 6.2) holds with high probability over the randomness of $S \sim \mathcal{D}^n$ and random subset $J \subset [n]$ ($|J| = m$):

$$\mathcal{R}(W_T, \mathcal{D}) \leq c_0 \mathcal{R}(W_T, S_{[n] \setminus J}) + O\left( \frac{1}{n-m} + \frac{1}{(n-m)m} \mathbb{E}\left[ \sum_{t=1}^{T} \frac{\gamma_t^2}{\sigma_t^2} L(W_{t-1})^2 \right] \right),$$

where $L(W_{t-1}) := \max_{z \in S} \|f(W_{t-1}, z)\|$ is the longest gradient norm of any training sample in $S$ at step $t$ and $m$ is the size of $J$. Since $W_T$ is independent of the index set $J$, the first term $\mathcal{R}(W_T, S_{[n] \setminus J})$ is upper bounded by $\mathcal{R}(W_T, S) + O(\frac{1}{\sqrt{n-m}})$ with high probability, using standard Hoeffding's inequality. By setting $m = n/2$, our generalization bound has an $O(\frac{1}{\sqrt{n}} + \frac{1}{n} + \frac{T}{n^2})$ rate. The new $1/n^2$ rate is obtained using the concentration of the difference between the gradient of training samples and that of the prior (See Lemma 6.1).

We also prove a high probability generalization bound for Stochastic Gradient Langevin Dynamics (SGLD) (see Theorem 6.3):

$$\mathcal{R}(W_T, \mathcal{D}) \leq c_0 \mathcal{R}(W_T, S_{[n] \setminus J}) + O\left( \frac{1}{n-m} + \frac{1}{n-m}\left(\frac{1}{b} + \frac{1}{m}\right) \mathbb{E}\left[ \sum_{t=1}^{T} \frac{\gamma_t^2}{\sigma_t^2} L(W_{t-1})^2 \right] \right).$$

We compare our bounds with other GLD/SGLD bounds obtained in [Mou et al., 2018, Negrea et al., 2019, Li et al., 2020] and the details can be found in Appendix B.

**CLD.** Using the PAC-Bayesian framework, we obtain a new generalization bound for Continuous Langevin Dynamics (CLD), defined by the stochastic differential equation $dW_t = -\nabla F(W_t, S)\, dt + \sqrt{2\beta^{-1}}\, dB_t$. The main term of the generalization bound scales as $O(1/n^2)$ (by choosing $m = n/2$) and does not grow to infinity as the training time $T$ increases. See Theorem G.6 for the details.

## 2 Other Related Work

**Stochastic Langevin Dynamics** Stochastic Langevin dynamics is a popular sampling and optimization method in machine learning [Welling and Teh, 2011]. Zhang et al. [2017], Chen et al. [2020] show a polynomial hitting time (hitting a stationary point) of SGLD in general non-convex setting. Raginsky et al. [2017] study the generalization and excess risk of SGLD in nonconvex settings and their bound depends inversely polynomially on a certain spectral gap parameter, which may be exponential small in the dimension. Continuous Langevin dynamics (SDE) with various noise structure has also been used extensively as approximations of SGD in literature (see e.g., [Li et al., 2017, 2021]). However, in terms of generalization, isotropic Gaussian noise is not a good approximation of the discrete noise in SGD (Zhu et al. [2019]).

**Nonvacuous PAC-Bayesian Generalization Bounds.** Dziugaite and Roy [2017] first present a non-vacuous PAC-Bayesian generalization bound on MNIST (0.161 for a 1-layer MLP, see column T-600 of Table 1 in their paper). They use a very different training algorithm that explicitly optimizes the PAC-Bayesian bound and the output distribution is a multivariate normal distribution. To computing the closed form of KL, they choose a zero-mean Gaussian distribution as the prior distribution. Zhou et al. [2019] obtain the first non-vacuous generalization bound for ImageNet via a different method. Their method does not require any continuous noise injected but assumes that the network can be significantly compressed (so that the prior distribution is supported over the set of discrete parameters with finite precision). To our best knowledge, it is the only work that utilizes a discrete prior for proving generalization bounds of deep neural networks. Our result for FGD/FSGD has a similar flavor in a high level, that is the optimization method has a finite precision. However, our results do not need any assumption on compressibility of the model and can be applied to nonconvex learning problems other than neural networks.

**Generalization bounds via Information theory.** Raginsky et al. [2017] first show that the expected generalization error $\mathbb{E}_{S \sim \mathcal{D}^n}[\mathcal{R}(W, \mathcal{D}) - \mathcal{R}(W, S)]$ is bounded by $\sqrt{2I(S; W)/n}$, where $I(S; W) :=$

$\mathrm{KL}\left(P(S,W)\,\middle\|\,P(S)\otimes P(W)\right)$ is the mutual information between the data set $S$ and the parameter $W$. This work motivates several subsequent studies [Pensia et al., 2018, Negrea et al., 2019, Bu et al., 2020, Wang et al., 2021]. The main goal in this line of work is to obtain a tight bound on the mutual information $I(S;W)$. This is again reduced to bounding the KL divergence and thus typically requires continuous injected noise (e.g., Wang et al. [2021], Negrea et al. [2019]).

## 3 Preliminaries

**Notations.** We assume that the training dataset $S = (z_1, .., z_n)$ is sampled from $\mathcal{D}^n$, where $\mathcal{D}$ is the population distribution over the data domain $\Omega$. The model parameter $w$ is in $\mathbb{R}^d$. The risk function $\mathcal{R} : \mathbb{R}^d \times \Omega \to [0,1]$ measures the error of a model on a datapoint. The loss function $f : \mathbb{R}^d \times \Omega \to \mathbb{R}$ is a proxy of the risk. The optimization algorithm minimizes the loss function and we assume we can compute the gradient of the loss function. We note that the loss function may be different from the risk function (e.g., 0/1 risk vs the cross-entropy loss). The empirical risk is $\mathcal{R}(w, S) = \frac{1}{|S|} \sum_{z \in S} \mathcal{R}(w, z)$ and population risk is $\mathcal{R}(w, \mathcal{D}) = \mathbb{E}_{z \sim \mathcal{D}}[\mathcal{R}(w, z)]$. Similarly, we can define the empirical loss $f(w, S)$ and population loss $f(w, \mathcal{D})$. For any $J = (j_1, .., j_m)$, we use $S_J$ to denote the sequence $(S_{j_1}, ..., S_{j_m})$. The subsequence $(A_i, A_{i+1}, ..., A_j)$ is denoted by $A_i^j$. We use $(A_1^n, B_1^m)$ to denote the merged sequence $(A_1, A_2, ..., A_n, B_1, ..., B_m)$. When the elements in sequence $J$ are distinct, we also use $J$ to represent the set consisting of all of its elements. We may also slightly abuse the notation of a random variable to denote its distribution. For example, $\mathbb{E}_{x \sim X}[f(x)]$ is a shorthand for $\mathbb{E}_{x \sim P_X}[f(x)]$, and $\mathrm{KL}\left(X \,\middle\|\, Y\right)$ means $\mathrm{KL}\left(P_X \,\middle\|\, P_Y\right)$. For a random variable $W$, we define $\mathcal{R}(W, S) = \mathbb{E}_{w \sim W}[\mathcal{R}(w, S)]$ and $\mathcal{R}(W, \mathcal{D}) = \mathbb{E}_{w \sim W}[\mathcal{R}(w, \mathcal{D})]$. The set $\{1, 2, ..., n\}$ is denoted by $[n]$.

**KL-divergence.** Let $P$ and $Q$ be two probability distributions. The Kullback–Leibler divergence $\mathrm{KL}\left(P \,\middle\|\, Q\right)$ is defined only when $P$ is absolute continuous with respect to $Q$ (i.e., for any $x$, $Q(x) = 0$ implies $P(x) = 0$). In particular, if $P$ and $Q$ are discrete distributions, then $\mathrm{KL}\left(P \,\middle\|\, Q\right) = \sum_x P(x) \ln \frac{P(x)}{Q(x)}$. Otherwise, if $P$ and $Q$ are continuous distributions, it is defined as $\int P(x) \ln \frac{P(x)}{Q(x)} \, dx$. The following Lemma 3.1 is frequently used in this paper and is a well known property of KL divergence (see Cover [1999, Theorem 2.5.3], Li et al. [2020], Negrea et al. [2019]).

**Lemma 3.1** (Chain Rule of KL). *We are given two random sequences $W = (W_0, ..., W_T)$ and $W' = (W'_0, ..., W'_T)$. Then, the following equation holds (given all KLs are well defined):*

$$\mathrm{KL}\left(W \,\middle\|\, W'\right) = \mathrm{KL}\left(W_0 \,\middle\|\, W'_0\right) + \sum_{t=1}^{T} \mathop{\mathbb{E}}_{w \sim W_0^{t-1}} \left[ \mathrm{KL}\left(W_t | W_0^{t-1} = w \,\middle\|\, W'_t | {W'}_0^{t-1} = w\right) \right].$$

*Here $W_t | W_0^{t-1} = w$ denotes the distribution of $W_t$ conditioning on $W_0^{t-1} = (W_0, \ldots, W_{t-1}) = w$.*

**PAC-Bayesian.** In this paper, we use the PAC-Bayesian bound presented in Catoni [2007] which enjoys a tighter $O(\mathrm{KL}\left(Q \,\middle\|\, P\right)/n)$ rate comparing to the traditional $O(\sqrt{\mathrm{KL}\left(Q \,\middle\|\, P\right)/n})$ bound, but with a slightly larger constant factor on the empirical error. We restate their bound as follows.

**Lemma 3.2** (Catoni's Bound). *(see e.g., Lever et al. [2013]) For any prior distribution $P$ independent of the training set $S$, any $\delta \in (0,1)$, and any $\eta > 0$, the following bound holds w.p. $\geq 1 - \delta$ over $S \sim \mathcal{D}^n$:*

$$\mathop{\mathbb{E}}_{W \sim Q}[\mathcal{R}(W, \mathcal{D})] \leq \eta C_\eta \mathop{\mathbb{E}}_{W \sim Q}[\mathcal{R}(W, S)] + C_\eta \cdot \frac{\mathrm{KL}\left(Q \,\middle\|\, P\right) + \ln(1/\delta)}{n} \quad (\forall Q), \qquad (1)$$

*where $C_\eta = \frac{1}{1 - e^{-\eta}}$ is an absolute constant.*

**Concentration inequality.** We use the following variant of McDiramid inequality (Lemma 3.3) to prove the concentration of cumulative gradient difference in Section 6. The proof is deferred to Appendix C.

**Lemma 3.3.** *Suppose $\Phi : [n]^m \to \mathbb{R}^+$ is order-independent[1] and $|\Phi(J) - \Phi(J')| \leq c$ holds for any adjacent $J, J' \in [n]^m$ satisfying $|J \cap J'| = m - 1$[2]. Let $J$ be $m$ indices sampled uniformly from $[n]$ without replacement. Then $\Pr_J \left[\Phi(J) - \mathbb{E}_J[\Phi(J)] > \epsilon\right] \leq \exp(\frac{-2\epsilon^2}{mc^2})$.*

---

[1] $\Phi(j_1, ..., j_m) = \Phi(j_{\pi_1}, ..., j_{\pi_m})$ holds for any input $J = (j_1, ..., j_m) \in \Omega^m$ and any permutation $\pi \in \mathbb{S}_m$.
[2] $J \cap J' := \{i \in [n] : i \in J \wedge i \in J'\}$.

## 4 Data-Dependent PAC-Bayesian Bound

The dominating term in the PAC-Bayesian bound (1) is $\mathrm{KL}\left(Q\,\middle\|\,P\right)/n$, where $P$ is a prior distribution independent of the training dataset $S$. Typically, without knowing any information from $S$, the best possible bound for $\mathrm{KL}\left(Q\,\middle\|\,P\right)$ we can hope is at least $\Theta(1)$ (it should not be a function of $n$ hence should not decrease with $n$). However, if we are allowed to see $m$ data points from $S$ when constructing our prior, we may produce better prediction on posterior $Q_S$. The following theorem enables us to use data-dependent prior in PAC-Bayesian bound. The proof is almost the same as Cantoni's original proof and we provide a proof for completeness in Appendix D.

**Theorem 4.1** (Data-Dependent PAC-Bayesian). *Suppose $J$ is a random sequence including $m$ indices uniformly sampled from $[n]$ without replacement. For any $\delta \in (0,1)$ and $\eta > 0$, we have w.p. $\geq 1 - \delta$ over $S \sim \mathcal{D}^n$ and $J$:*

$$\mathcal{R}(Q,\mathcal{D}) \leq \eta C_\eta \mathcal{R}(Q,S_I) + C_\eta \cdot \frac{\mathrm{KL}\left(Q\,\middle\|\,P(S_J)\right) + \ln(1/\delta)}{n-m} \quad (\forall Q),$$

*where $I = [n]\backslash J$ is the set of indices not in $J$, $P(S_J)$ is the prior distribution only depending on the information of $S_J$ ($S_J$ is the subset of $S$ indexed by $J$), and $C_\eta := \frac{1}{1-e^{-\eta}}$ is a constant.*

**Remarks.** Note that the above bound holds regardless of whether $Q$ depends on $S$ or not. Also note that the first term in the right hand side is $\mathcal{R}(Q,S_I)$, not $\mathcal{R}(Q,S)$ as in the usual generalization bounds. We remark that for most of our learning algorithms that are independent of $J$ (i.e., changing $J$ does not change the output $Q$), by standard Chernoff-Hoeffding inequality, $\mathcal{R}(Q,S_I)$ can be bounded by $\mathcal{R}(Q,S) + O(1/\sqrt{n-m})$ with high probability over the randomness of $J$. For example, the update rules of GLD, SGLD and CLD are independent of $J$, hence $\mathcal{R}(Q,S_I)$ can be replaced by $\mathcal{R}(Q,S) + O(1/\sqrt{n-m})$ in Theorem 4.1. However, we point out a subtle point that FGD (Algorithm 1) studied in this paper depends on $J$. It may be the case that by knowing $J$, FGD extracts more information from $S_J$ but not much from $S_I$, unintentionally making $\mathcal{R}(Q,S_I)$ a validation error, rather than the training error as it should be. However, from our experiment (see Figure 5 and 6 in Appendix H, and Figure 2a), we can see that FGD is very close to GD and the $S_I$ error $\mathcal{R}(W_T,S_I)$ is indeed close to the training error $\mathcal{R}(W_T,S)$ and both are significantly smaller than the testing error $\mathcal{R}(W_T,\mathcal{D})$. So $\mathcal{R}(Q,S_I)$ can be considered as a genuine training error in our study of FGD.

## 5 FGD and FSGD

In this section, we study the generalization error of finite precision variants of gradient descent and stochastic gradient descent: Floored Gradient Descent (FGD) and Floored Stochastic Gradient Descent (FSGD).

First we need to define the "floor" operation which is used in the definitions of FGD and FSGD.

**Definition 5.1** (Floor). *For any vector $X \in \mathbb{R}^d$, let $Y = \mathrm{floor}(X)$ defined as:*

$$Y_i = \mathrm{floor}(X_i) = \lfloor X_i \rfloor \text{ if } X_i \geq 0, \quad = -\lfloor -X_i \rfloor \text{ if } X_i < 0, \text{ for all } i \in [d].$$

**FGD:** The Floored Gradient Descent algorithm is formally defined in Algorithm 1, where $(\gamma_t)_{t\geq 0}$ and $(\varepsilon_t)_{t\geq 0}$ are the step size and precision sequences, respectively. For a subset $Z \subseteq S$, we write $\nabla f(W_{t-1},Z) := \frac{1}{|Z|}\sum_{z\in Z}\nabla f(W_{t-1},z)$. Note that FGD can be viewed as gradient descent with given precision limit $\varepsilon_t$. We can see if we ignore the floor operation or let $\varepsilon_t$ approach 0, FGD reduces to the ordinary GD (see Appendix A). We also study momentum FGD, in which the 5th line of Algorithm 1 is replaced by

$$W_t \leftarrow W_{t-1} + \alpha \cdot (W_{t-1} - W_{t-2}) - g_2 - \varepsilon_t \cdot \mathrm{floor}((g_1 - g_2)/\varepsilon_t);$$

Here $\alpha > 0$ is a constant. We remark that both FGD and its momentum version are deterministic algorithms. The following theorem provides the generalization error bound for both algorithms.

**Theorem 5.2.** *Suppose $J$ is a random sequence consisting of $m$ indices uniformly sampled from $[n]$ without replacement. Then for any $\delta \in (0,1)$, both FGD (Algorithm 1) and its momentum version satisfy the following generalization bound w.p. at least $1 - \delta$ over $S \sim \mathcal{D}^n$ and $J$:*

$$\mathcal{R}(W_T,\mathcal{D}) \leq \eta C_\eta \mathcal{R}(W_T,S_I) + C_\eta \cdot \frac{\ln(1/\delta)+3}{n-m} + \frac{C_\eta \ln(dT)}{n-m}\sum_{t=1}^{T}\left(\frac{\gamma_t^2}{\varepsilon_t^2}\|\mathbf{g}_t\|^2\right),$$

---

**Algorithm 1:** Floored Gradient Descent (FGD)

---

**Input:** Training dataset $S = (z_1, .., z_n)$. Index set $J$.

**Result:** Parameter $W_T \in \mathbb{R}^d$.

1   Initialize $W_0 \leftarrow w_0$;

2   **for** $t : 1 \to T$ **do**

3      $g_1 \leftarrow \gamma_t \nabla f(W_{t-1}, S)$;

4      $g_2 \leftarrow \gamma_t \nabla f(W_{t-1}, S_J)$;

5      $W_t \leftarrow W_{t-1} - g_2 - \varepsilon_t \cdot \text{floor}((g_1 - g_2)/\varepsilon_t)$;

6   **end**

---

*where $d$ is the dimension of parameter space, $I = [n] \backslash J$ is the set of indices not in $J$, $C_\eta := \frac{1}{1-e^{-\eta}}$ is a constant, and $\mathbf{g}_t := \nabla f(W_{t-1}, S) - \nabla f(W_{t-1}, S_J)$.*

*Proof.* We use Theorem 4.1 to prove our theorem for the momentum version. The ordinary FGD is a special case of the momentum version with $\alpha = 0$. The key is to construct the prior distribution $P(S_J)$ such that $\text{KL}\left(W_T \,\|\, P(S_J)\right)$ is tractable. Let $p$ be any real number in $(0, 1/3)$. We first define a stochastic process $\{W'_0, \ldots, W'_T\}$, by the following update rule ($W'_0 := w_0$):

$$W'_t \leftarrow W'_{t-1} + \alpha \cdot \left(W'_{t-1} - W'_{t-2}\right) - \gamma_t \nabla f(W'_{t-1}, S_J) - \varepsilon_t \cdot \xi_t,$$

where $\xi_t$ is a discrete random variable such that for all $(a_1, .., a_d) \in \mathbb{Z}^d$:

$$\Pr[\xi_t = (a_1, ..., a_d)^\top] := \left( \sum_{i=-\infty}^{\infty} p^{i^2} \right)^{-d} \exp\left( -\sum_{k=1}^{d} \ln(1/p) a_k^2 \right).$$

It is easy to verify that the sum of the probabilities ($\sum_{a \in \mathbb{Z}^d} \Pr[\xi_t = a]$) equals to 1. Note that $W'_t$ only depends on $S_J$. We define $P(S_J)$ as the distribution of $W'_T$.

Recall that $W_0^t = (W_0, ..., W_t)$ is the parameter sequence of FGD (Algorithm 1). Applying the chain rule of KL-divergence (Lemma 3.1), we have:

$$
\begin{aligned}
\text{KL}\left(W_T \,\|\, P(S_J)\right) = \text{KL}\left(W_T \,\|\, W'_T\right) &\leq \text{KL}\left(W_0^T \,\|\, {W'}_0^T\right) \\
&= \sum_{t=1}^{T} \mathop{\mathbb{E}}_{w \sim W_0^{t-1}} \left[ \text{KL}\left(W_t | W_0^{t-1} = w \,\|\, W'_t | {W'}_0^{t-1} = w\right) \right] \qquad (2) \\
&= \sum_{t=1}^{T} \text{KL}\left(W_t | W_0^{t-1} = W_0^{t-1} \,\|\, W'_t | {W'}_0^{t-1} = W_0^{t-1}\right).
\end{aligned}
$$

The last equation holds because FGD is deterministic. Let $w = W_0^{t-1}$. The distribution of $W_t | W_0^{t-1} = w$ (where $w = (w_0, ..., w_{t-1})$) is a point mass on

$$w_{t-1} + \alpha \cdot (w_{t-1} - w_{t-2}) - \gamma_t \nabla f(w_{t-1}, S_J) - \varepsilon_t \cdot \text{floor}\left( \frac{\gamma_t(\nabla f(w_{t-1}, S) - \nabla f(w_{t-1}, S_J))}{\varepsilon_t} \right).$$

Let vector $a = (a_1, \ldots, a_d) = \text{floor}(\frac{\gamma_t}{\varepsilon_t}(\nabla f(w_{t-1}, S) - \nabla f(w_{t-1}, S_J)))$. By the definition of $W'_t$, we have

$$
\begin{aligned}
\text{KL}\left(W_t | W_0^{t-1} = w \,\|\, W'_t | {W'}_0^{t-1} = w\right) &= 1 \cdot \ln\left(1/ \Pr\left[\xi_t = a\right]\right) \\
&= \ln\left( \left( \sum_{i=-\infty}^{\infty} p^{i^2} \right)^d \right) + \sum_{k=1}^{d} \ln(1/p) \cdot a_k^2.
\end{aligned}
$$

Since $|i| \leq i^2$ and $p \in (0, 1/3)$, we have $\ln\left( (\sum_{i=-\infty}^{\infty} p^{i^2})^d \right)$ is at most $d \ln\left(1 + 2 \sum_{i=1}^{\infty} p^i\right)$. It can be further bounded by $d \ln(1 + 3p)$. Moreover, it can be bounded by $3dp$ as $\ln(1 + x) \leq x$. Thus, the above KL-divergence can be bounded by $3dp + \sum_{k=1}^{d} \ln(1/p) a_k^2$. Recall that the

$k$th entry of $a$ is $a_k := \lfloor \frac{\gamma_t}{\varepsilon_t} \cdot (\nabla_k f(w_{t-1}, S) - \nabla_k f(w_{t-1}, S_J)) \rfloor$, which is less than or equal to $\frac{\gamma_t}{\varepsilon_t} \cdot (\nabla_k f(w_{t-1}, S) - \nabla_k f(w_{t-1}, S_J))$. Therefore, we have

$$\mathrm{KL}\left(W_t | W_0^{t-1} = w \,\big|\big|\, W_t' | W_0'^{t-1} = w\right) \le 3dp + \frac{\ln(1/p)\gamma_t^2}{\varepsilon_t^2} \|\nabla f(w_{t-1}, S) - \nabla f(w_{t-1}, S_J)\|_2^2.$$

Plugging the above inequality into (2), we have

$$\mathrm{KL}\left(W_T \,\big|\big|\, P(S_J)\right) \le \sum_{t=1}^{T} \left(3dp + \frac{\ln(1/p)\gamma_t^2}{\varepsilon_t^2} \|\nabla f(W_{t-1}, S) - \nabla f(W_{t-1}, S_J)\|_2^2\right).$$

We conclude our proof by plugging it into Theorem 4.1 (setting $p = 1/(Td)$). $\qquad\square$

**FSGD:** We can use a similar approach to prove a generalization bound for Floored Stochastic Gradient Desent (FSGD). Formally, FSGD is identical to Algorithm 1 except for the definitions of $g_1$ and $g_2$ replaced with:

$$g_1 \leftarrow \nabla f(W_{t-1}, S_{B_t}), \quad g_2 \leftarrow \nabla f(W_{t-1}, S_{B_t \cap J}),$$

where $B_t \subseteq [n]$ is a random batch independent of $S, J$ and $W_0^{t-1}$. Formally, each $B_t$ is a set including $b$ indices uniformly sampled from $[n]$ without replacement. The following theorem provides a generalization bound for FSGD. The proof can be found in Appendix E.

**Theorem 5.3.** *Suppose $J$ is a random sequence consisting of $m$ indices uniformly sampled from $[n]$ without replacement. Then for any $\delta \in (0,1), \varepsilon \in (0,1)$, FSGD satisfies the following generalization bound: w.p. at least $1 - \delta$ over $S \sim \mathcal{D}^n$ and $J$:*

$$\mathcal{R}(W_T, \mathcal{D}) \le \eta C_\eta \mathcal{R}(W_T, S_I) + C_\eta \cdot \frac{\ln(1/\delta) + 3}{n - m} + \frac{C_\eta \ln(dT)}{n - m} \mathop{\mathbb{E}}_{B_0^T} \left[\sum_{t=1}^{T} \frac{\gamma_t^2}{\varepsilon_t^2} \|\mathbf{g}_t\|^2\right],$$

*where $d$ is the dimension of parameter space, $I = [n]\backslash J$, $C_\eta := \frac{1}{1 - e^{-\eta}}$ is a constant, and $\mathbf{g}_t := f(W_{t-1}, S_{B_t}) - \nabla f(W_{t-1}, S_{J \cap B_t})$.*

## 6 Gradient Langevin Dynamics

In this section, we present new generalization bounds for Gradient Langevin Dynamics (GLD) and Stochastic Gradient Langevin Dynamics (SGLD) based on Theorem 4.1.

**Gradient Langevin Dynamics (GLD):** The GLD algorithm can be viewed as gradient descent plus a Gaussian noise. Formally, for a given training set $S \sim \mathcal{D}^n$, the update rule of GLD is defined as follows:

$$W_{t+1} \leftarrow W_t - \frac{\gamma_{t+1}}{n} \sum_{z \in S} \nabla f(W_t, z) + \sigma_{t+1} \mathcal{N}(0, I_d), \tag{GLD}$$

Here the gradient $\nabla f(W_t, z)$ can be replaced with any gradient-like vector such as a clipped gradient. The output of GLD is the last step parameter $W_T$ or some function of the whole training trajectory $W_0^T$ (e.g., the average of the suffix $\frac{1}{K} \sum_{t=T-K}^{T} W_t$).

We still use the data-dependent PAC-Bayesian framework (Theorem 4.1) to prove the generalization bound for GLD. Unlike FGD (Algorithm 1), GLD is independent of the prior indices $J$, which enables us to prove the following concentration bound (Lemma 6.1) for the gradient difference. The proof is based on Lemma 3.3, which is postponed to Appendix F.

**Lemma 6.1.** *Let $S = (z_1, ...z_n)$ be any fixed training set. $J$ is a random sequence including $m$ indices uniformly sampled from $[n]$ without replacement, and $W = (W_0, ..., W_T)$ is any random sequence independent of $J$. Then the following bound holds with probability at least $1 - \delta$ over the randomness of $J$:*

$$\mathop{\mathbb{E}}_{W} \left[\sum_{t=1}^{T} \frac{\gamma_t^2}{\sigma_t^2} \|\nabla f(W_{t-1}, S) - \nabla f(W_{t-1}, S_J)\|^2\right] \le \frac{C_\delta}{m} \mathop{\mathbb{E}}_{W} \left[\sum_{t=1}^{T} \frac{\gamma_t^2}{\sigma_t^2} L(W_{t-1})^2\right],$$

*where $C_\delta = 4 + 2\ln(1/\delta) + 5.66\sqrt{\ln(1/\delta)}$, and $L(w) = \max_{i \in [n]} \|\nabla f(w, z_i)\|$.*

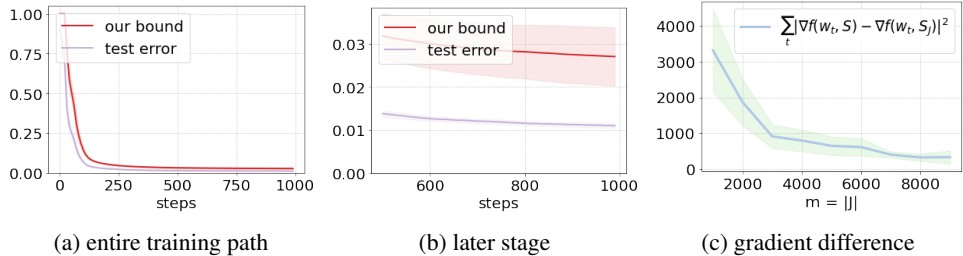

(a) entire training path      (b) later stage      (c) gradient difference

Figure 1: MNIST + CNN + FGD. In (a) and (b), we plot the true test error and our bound (Theorem 5.2 with $\eta = 1.5, \delta = 0.1$). In (c), we show how cumulative gradient difference decreases as $m$ (the size of $J$) increases.

Now we are ready to present our main results. The proofs can be found in Appendix F.

**Theorem 6.2.** *Suppose $J$ is a random sequence consisting of $m$ indices uniformly sampled from $[n]$ without replacement. Let $W_T$ be the output of GLD. Then for any $\delta \in (0, \frac{1}{2})$ and $\eta > 0$, we have w.p. $\geq 1 - 2\delta$ over $S \sim \mathcal{D}^n$ and $J$, the following holds ($L(w) := \max_{z \in S} \|f(w, z)\|$):*

$$\mathcal{R}(W_T, \mathcal{D}) \leq \eta C_\eta \mathcal{R}(W_T, S_I) + \frac{C_\eta \ln(1/\delta)}{n - m} + \frac{C_\eta C_\delta}{2(n - m)m} \underset{W_0^T}{\mathbb{E}} \left[ \sum_{t=1}^T \frac{\gamma_t^2}{\sigma_t^2} L(W_{t-1})^2 \right],$$

*where $C_\delta = 4 + 2\ln(1/\delta) + 5.66\sqrt{\ln(1/\delta)}$, $I = [n] \backslash J$ and $C_\eta = \frac{1}{1 - e^{-\eta}}$.*

**Stochastic Gradient Langevin Dynamics (SGLD):** For a given training data set $S$, the update rule of SGLD is defined as:

$$W_{t+1} \leftarrow W_t - \gamma_{t+1} \nabla f(W_t, S_{B_t}) + \sigma_{t+1} \mathcal{N}(0, I_d), \tag{SGLD}$$

where $B_t \sim \text{uniform}([n])^b$ is the mini-batch of size $b$ at step $t$. Note that $B_t$ is a sequence instead of a set, thus it may include duplicate elements. Similar to the analysis of GLD, we can prove the following bound for SGLD.

**Theorem 6.3.** *Let $W_T$ be the output of SGLD when the training set is $S$, and $J$ be a random sequence with $m$ indices uniformly sampled from $[n]$ without replacement. For any $\delta \in (0, 1)$ and $m \geq 1$, we have w.p. $\geq 1 - 2\delta$ over $S \sim \mathcal{D}^n$ and $J$, the following holds:*

$$\mathcal{R}(W_T, \mathcal{D}) \leq \eta C_\eta \mathcal{R}(W_T, S_I) + \frac{C_\eta \ln(1/\delta)}{n - m} + \frac{C_\eta}{n - m} \left( \frac{4}{b} + \frac{C_\delta}{2m} \right) \underset{W_0^T}{\mathbb{E}} \left[ \sum_{t=1}^T \frac{\gamma_t^2}{\sigma_t^2} L(W_{t-1})^2 \right],$$

*where $L(w) := \max_{z \in S} \|f(w, z)\|$, $C_\delta = 4 + 2\ln(1/\delta) + 5.66\sqrt{\ln(1/\delta)}$, $C_\eta = \frac{1}{1 - e^{-\eta}}$, $b$ is the batch size, and $I = [n] \backslash J$.*

**Remark 6.4.** *If the gradient norm is bounded[3] and we use a decaying learning rate schedule such as $\gamma_t \propto O(1/t)$, then the summation in our bound converges. Hence, under such a learning rate schedule, Theorem 6.2 and 6.3 imply the following test error bound for GLD or SGLD: $\mathcal{R}(W_T, \mathcal{D}) \leq \eta C_\eta \mathcal{R}(W_T, S_I) + \widetilde{O}(\frac{1}{n-m})$ which is independent of $T$, where $\widetilde{O}$ hides some logarithmic factors.*

## 7 Experiment

In this section, we conduct experiments for FGD and FSGD on MNIST [LeCun et al., 1998] and CIFAR10 [Krizhevsky et al., 2009] to investigate the the optimization and generalization properties of FGD and FSGD, and the numerical closeness between our theoretical bounds and true test errors. Due to space limit, the detailed experimental setting and some additional experimental results can be found in Appendix H.

**FGD/FSGD vs GD/SGD.** We first demonstrate that the training and testing curves of FGD and GD are nearly identical (we choose precision level $\varepsilon = 0.005$ or $0.004$). We also show that the same is true for FSGD vs SGD. Due to space limit, the figures are presented in Appendix H (Figure 5 and 6).

---

[3] $\|\nabla f(w, z)\| \leq L$ holds for all $w, z$

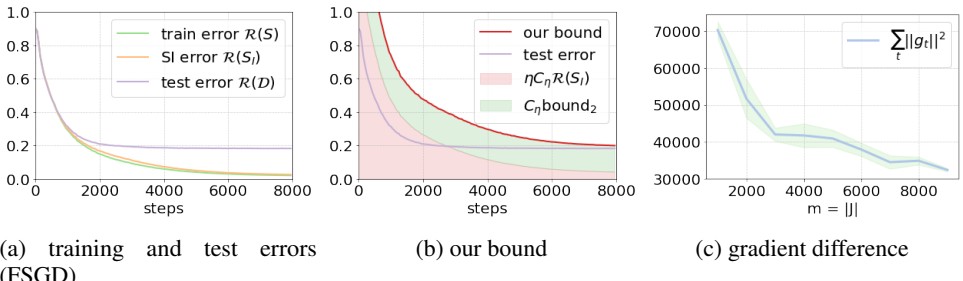

(a) training and test errors (FSGD)    (b) our bound    (c) gradient difference

Figure 2: CIFAR10 + SimpleNet + FSGD. In (a), we plot $\mathcal{R}(W_T, S_I)$, $\mathcal{R}(W_T, S)$ and the test error. We can see that $\mathcal{R}(W_T, S_I)$ is very close to $\mathcal{R}(W_T, S)$. In (b), we plot our theoretical bound (Theorem 5.3 with $\eta = 2, \delta = 0.1$). The red part corresponds to the first term of our bound (the empirical risk) and the green part corresponds to the rest. The last step test error and our bound are $0.18$ and **0.198**, respectively. In (c), we show how cumulative gradient difference decreases as $m$ (the size of $J$) increases.

**Non-vacuous bounds.** For MNIST, we train a CNN ($d = 1.4 \cdot 10^6$) by FGD with $\gamma_t = 0.005 \cdot 0.9^{\lfloor \frac{t}{150} \rfloor}$ and $\varepsilon_t = 0.005$ and momentum $\alpha = 0.9$). The size $m = |J|$ is set to $n/2 = 30000$. As shown in Figure 1a and 1b, our bound (Theorem 5.2 with $\eta = 1.5, \delta = 0.1$) tracks the testing error closely. At step $T = 990$, our bound is **0.026** while the testing error is $0.011$. This is non-vacuous and tighter than best known $11\%$ MNIST bound reported in Dziugaite et al. [2021]. For CIFAR10, we train a SimpleNet [Hasanpour et al., 2016] without BatchNorm and Dropout. The number of parameters $d$ is nearly $18 \cdot 10^6$. We use FSGD to train our model. The learning rate $\gamma_t$ is set to $0.001 \cdot 0.9^{\lfloor t/200 \rfloor}$, the precision $\varepsilon_t$ is set to $0.004$, and the momentum $\alpha$ is set to $0.99$. The batch size is 2000. $m = |J|$ is set to $n/5 = 10000$. The result is shown in Figure 2b. We stop training at step $t = 8000$ when the testing error is and $0.18$. At that time, our testing error bound is **0.198** which is non-vacuous and tighter than best known $0.23$ CIFAR10 bound reported in Dziugaite et al. [2021].

**Decrease of the gradient difference.** Intuitively, the cumulative squared norm of gradient difference $\mathbf{g}_t := \nabla f(W_t, S) - \nabla f(W_t, S_J)$ should decrease as $m = |J|$ increases. Although we cannot prove a concentration like Lemma 6.1 (i.e., $\|\mathbf{g}_t\|^2$ scales as $O(1/m)$), we can still observe that $\|\mathbf{g}_t\|^2$ decreases when $m$ increases. The results are depicted in Figure 1c and Figure 2c.

**Random labels.** We conduct the random label experiment designed in Zhang et al. [2017]. Our theoretical bounds can distinguish the datasets with different portion ($p$) of random labels. See Appendix H.

# 8  Conclusion

In this paper, we prove new generalization bounds for several gradient-based methods with either discrete or continuous noises based on carefully constructed data-dependent priors. Recall that FGD requires to compute the gradient difference for technical reasons. It would be more natural and desirable if we only need to compute the full gradient and rounded to the nearest grid point. An intriguing future direction is to free FGD/FSGD from the dependence of the prior subset $J$ so that we can apply the concentration on the gradient difference to obtain a tighter bound. Of course, a major further direction is to obtain similar generalization bounds for vanilla GD and SGD, which remains to be an important open problem in this line of work. Our technique can be useful for handling deterministic algorithms and discrete noises, but it seems that new technical ideas or assumptions are needed for tackling GD or SGD.

# 9  Acknowledgements

The authors would like to thank the anonymous reviewers for their constructive comments. The authors are supported in part by the National Natural Science Foundation of China Grant 62161146004, Turing AI Institute of Nanjing and Xi'an Institute for Interdisciplinary Information Core Technology.

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
