# OpenReview forum: "Generalization Bounds for Gradient Methods via Discrete and Continuous Prior"
_NeurIPS.cc/2022/Conference — NeurIPS 2022 Accept_

### Official Review · Reviewer_4EBj · 2022-07-06

**Rating:** 7
**Confidence:** 4
**Soundness:** 4 excellent
**Presentation:** 3 good
**Contribution:** 4 excellent

**Summary:**

This paper utilizes data-dependent PAC-Bayesian bounds to derive generalization errors for a variety of gradient-based learning algorithms. The key idea is to construct a special prior based on a run of the algorithm from a randomly selected subset of the data. In the algorithms studied, these priors lead to tractble upper bounds on the PAC-Bayesian bounds, and thus bounds on the overall generalization error. In tests on MNIST, the bounds are quite tight, and they also give non-vacuous bounds on generalization error for CIFAR.

**Questions:**

From what I can tell, the FGD is deterministic once the data set is fixed. As a result, Theorem 5.2 can be applied to a single run on FGD to give a high-probability bound on the test error.

In contrast, it would seem that this is not necessarily possible for the stochastic algorithms, FSGD and the Langevin variants. In particular, the term $\mathcal{R}(W_T,S_I)$ requires averaging out the effects of the randomness of the algorithms. Is this the correct way to interpret this?

Based on that question above, is there a way to get true high-probability bounds for these stochastic algorithms? Would it just require running them several times (for a fixed dataset and selected $J$) and doing a concentration bound on the training error? Is there something more clever that can be done?

**Limitations:**

Yes. They indicate that the potential for negative societal impact is not applicable, and I agree.

**Strengths And Weaknesses:**

Strengths
The methodology for deriving the bounds is quite interesting, and the bounds themselves give nice insights into the performance of various algorithms. The basic ideas seem quite extensible to a variety of different algorithms. The paper is well-written and well-organized.

Weaknesses
The paper does have some typos and basic mistakes. All of them appear to be fixable.

Lemma 3.2 appears to be stated incorrectly. The bound on the right should probably include an empircal risk term, not a population risk term. However, this lemma is never actually used in the paper, from what I can tell. (The correctly stated Theorem 4.1 is used.) So, it could likely be removed.

There are a few fixable mistakes at the end of the proof of Theorem 4.1. Specifically, the equality between lines 517 and 518 does not hold. A related inequality, based on convexity does hold, but that goes the wrong way. So, a different approach is needed.

Edit: Based on discussions, some better motivation for the algorithms is required.

Here is a way to fix it. Let $c=\frac{\lambda}{n-m}$ and recall that $w$ is fixed. Note that for random $z_i$, $\mathcal{R}(w,z_i)$ is a Bernoulli random variable which has expected value $\mathcal{R}(w,\mathcal{D})$. For simpler notation, set $b=\mathcal{R}(w,z_i)$ and $q=\mathcal{R}(w,\mathcal{D})$. Direct calculation shows that
$$
E_b\left[e^{c(\Phi(q)-b)} \right] =
\frac{e^{0}(1-q) + e^{-c}q}{1-(1-e^{-c})q}=
1
$$

The reference to Theorem 1 on line 538 probably should refer to Theorem 5.2

In line 554, the bound should probably be $\|V_t\|\le \frac{2}{m} L(W_{t-1})$. Based on this, the looser bound from line 551 seems unnecessary.

---

> ### Author Response · Authors · 2022-08-02
> **Response to Reviewer 4EBj**
>
> Thanks for your encouraging and insightful comments! We have fixed the typos and addressed your question as follows.
>
> 1. ``Lemma 3.2 appears to be stated incorrectly.'':
> Response:
> Indeed, the $R(W,D)$ term in the RHS should be replaced with $R(W,S)$. We have corrected it in the revised version.
>
> 2. "There are a few fixable mistakes at the end of the proof of Theorem 4.1. '':
> Response:
> Thank you for pointing it out! Your elegant proof is helpful (which we will acknowledge) and we have updated the proof in the revised version (See Appendix C).
>
> 3. "the looser bound from line 551 seems unnecessary.'':
> Response: Thanks for pointing it out.
> Indeed, the sentence ``Note that $\left\|V_t\right\|$ is at most $2L(W_{t-1}) = 2 \max_{i\in[n]}\left\|\nabla f(W_{t-1},z_i)\right\|$'' can be removed.
>
> 4. "In particular, the term $R(W_T,S_I)$ requires averaging out the effects of the randomness of the algorithms. Is this the correct way to interpret this?'':
> Response:
> Yes. $R(W_T,S_I)$ is defined by $\mathrm{E}_{w\sim W_T}[R(w,S_I)]$ which takes the expectation over the randomness of learning algorithm.
>
> 5. "is there a way to get true high-probability bounds for these stochastic algorithms?''
> Response:
> It is a great question! It seems that this is difficult for PAC-Bayesian framework, and we are not aware of any prior work in this direction. We suspect that other methods such the complexity based method (e.g., covering number or Rademacher complexity framework) might work since our discrete methods (FGD/FSGD) has limited precision which may enjoy small model complexity. We do not have all details yet and it is an intriguing future direction.

---

> > ### Comment · Reviewer_4EBj · 2022-08-09
> > **Discussion**
> >
> > Thank you for your response. While I am still quite positive about this paper, several reviewers flagged issues about the relevance of the FGD algorithm. While I see your point that this algorithm has nice analytical properties, I also see the point of the other reviewers. As a result, I am lowering my score to a still strong 7.

---

### Official Review · Reviewer_XXfA · 2022-07-07

**Rating:** 5
**Confidence:** 2
**Soundness:** 3 good
**Presentation:** 2 fair
**Contribution:** 2 fair

**Summary:**

This paper is concerned with proving new generalization bounds for gradient descent methods, by using suitable priors. To this end, the Catoni's PAC-Bayesian framework is used. But, unlike other work, this paper does not use restrictive assumptions on the learning rate or continuous injected noise.
More precisely, the authors show a high-probability generalization bound for Floored gradient descent.
In a similar way, the authors derive generalization bounds for (stochastic) gradient Langevin dynamics.

**Questions:**

1. Floor Gradient descent seems a bit constructed. Why would one be interested in that? Why would one use a mini-batch gradient if one computes the full gradient anyway?
2. There are some references missing that also don't come with the restrictions of a step-size schedule or injected noise, e.g. https://proceedings.neurips.cc/paper/2020/hash/37693cfc748049e45d87b8c7d8b9aacd-Abstract.html
3. I would suggest to carefully clean up the text, there are quite a few language mistakes.

Overall, the clarity of the writing and presentation has to be improved. Like this, I cannot recommend the paper for NeurIPS.

**Limitations:**

The authors have adequately addressed the limitations. There are no potential negative societal impacts.

**Strengths And Weaknesses:**

Strengths: Generalization bounds are an extremely central topic of ML. As far as I know, these bounds are novel. While floored gradient descent seems a bit artificial, the extension to stochastic gradient Langevin dynamics is very useful. The authors explain suitably how they choose the prior P. The literature review is nice and useful.

Weaknesses:
--- The writing is unfortunately poor, many typos and formulations which are difficult to understand. Due to this, I found the paper lacking in quality and could not follow the flow of the arguments in many cases.
--- The main result is on floored gradient descent, which is not a very relevant method. According to Algorithm 1, it seems to consist of computing the full gradient and a batch gradient. Why would one compute the batch gradient if one already afforded to compute the full gradient? This has to be much better justified...
--- The concept of data-dependent prior needs more explanation. Is this a usual concept in the literature?
--- The experiment section is very short.
--- No plots or illustrations are given to provide intuition. It would, however, be important to give intuition for the data-dependent priors, etc.
--- The paper ends abruptly, without any conclusion or discussion of future work.

---

> ### Author Response · Authors · 2022-08-02
> **Response to Reviewer XXfA**
>
> Thanks for the helpful comments!
> In the revised version, we have fixed several typos (which we apologize),
> done several new experiments to better
> verify the effectiveness of our bounds and add several discussions to better motivate our results. We believe the revised version has improved significantly and hope you can take
> it into consideration.
>
>
> Our responses to specific questions are as follows.
>
> 1. Floor Gradient descent seems a bit constructed. Why would one be interested in that? :
> Thanks for the question. Please see our response titled "Why study FGD/FSGD".
>
> 2. Why would one use a mini-batch gradient if one computes the full gradient anyway?
> FGD is a finite precision variant of GD. It decomposes the full gradient $g_1:=\gamma_t \nabla f(W_t,S)$ (used in GD) into a sum of two parts, $g_2:=\gamma_t \nabla f(W_t, S_J)$ and $\Delta g:=\gamma_t (g_1-g_2)=\gamma_t\nabla f(W_t, S) - \gamma_t\nabla f(W_t,S_J)$, where $S_J$ is the ``prior data'' instead of a mini-batch. Then we reduce the precision of $\Delta g$ to $\varepsilon_t$ by applying a floor-operation $\Delta' g:=\varepsilon_t \mathrm{floor}(\Delta g/\varepsilon_t)$. After that, $g_2 + \Delta' g$ can be viewed as an approximation of full gradient $g_1=g_2 + \Delta g$. It is easy to see that when $\varepsilon_t$ goes to $0$, FGD becomes GD.
> Again we stress that the goal of introducing FGD is not to present a better optimization algorithm, but to use them to understand the generalization performance of GD/SGD and
> to extend the mathematical techniques for proving tighter generalization bounds for GD
> and SGD.
>
> 3. ``There are some references missing'': Thanks for pointing it out! [1] provides another angle to study the generalization error of SGD, namely the Hausdorff dimension $\mathrm{d_H}$, which is algorithm and data dependent. Their obtained generalization bound is $O(\sqrt{\frac{\mathrm{d_H}\log n + \log(1/\delta)}{n}})$. Their bound is not directly comparable to the trajectory-based bounds. We have included a discussion of this paper in the revised version.
>
> Reference
>
> [1] Simsekli, Umut, et al. "Hausdorff dimension, heavy tails, and generalization in neural networks." Advances in Neural Information Processing Systems 33 (2020): 5138-5151.

---

> > ### Comment · Reviewer_XXfA · 2022-08-03
> > **Thank you for your comments**
> >
> > Thank you for your comments, but my score remains unchanged!
> >
> > To make the paper acceptable, major rewritings of the material are necessary. In particular, (i) the language has to be improved, (ii) more experiments are needed, (iii) the motivation for floor gradient has to be better explained, (iv) the issue with computing the full gradient has to be better discussed.
> >
> > Thank your nonetheless for your hard work and best wishes for a revision!

---

> > > ### Author Response · Authors · 2022-08-06
> > > **Reply to "Thank you for your comments "**
> > >
> > > Thanks for your reply! We have made substantial changes and addressed the problems you mentioned in the revised version. For the convenience of your reading, we marked the major parts we modified in violet color in the revised version, and list them as follows:
> > >
> > > 1. Language: We have polished the entire draft and we believe most of the typos have been fixed (small typos were not colored).
> > >
> > > 2. Experiments: Our experiments has been significantly improved (partly in order to address another reviewer's comments). Our newly reported  MNIST and CIFAR10 test error bounds (0.0288 and 0.196) are already smaller than the state-of-the-art bounds (0.11 and 0.23). You can check this update in Section 5.1 and Appendix G.
> > >
> > > 3. Motivation for FGD: We have discussed the motivation in detail in another reply titled "Why study FGD/FSGD?", and we have added that discussion to the Section 1.2 (around line 120) of the main text. Although FGD/FSGD are not commonly used algorithms, we have verified that they perform very close to GD/SGD, hence useful for understanding vanilla GD/SGD. Further detailed motivations are provided in the revision and the response.
> > >
> > > 4. "Full gradient issue": We have explained it clearly in our last response and this is not an issue. We have also added this discussion in the beginning of Appendix D.
> > >
> > > Hope our response can address your concerns. Should you have more questions or specific points you think we have not addressed thoroughly, we are happy to answer them.

---

### Official Review · Reviewer_Zg1r · 2022-07-08

**Rating:** 5
**Confidence:** 3
**Soundness:** 3 good
**Presentation:** 2 fair
**Contribution:** 2 fair

**Summary:**

This paper studies the generalization property of gradient-based optimization algorithms using PAC-Bayesian bound. To this end, the authors first provide a bound using data-dependent prior (Theorem 4.1) and use it to prove generalization bounds for FGD, FSGD, GLD, and SGLD. The authors also empirically observed that some proposed bounds are not vacuous for the MNIST classification task.

**Questions:**

All questions are listed in the Strength and Weakness section.

**Limitations:**

The authors mentioned that the conclusion section (in Appendix) contains the limitation; however, I couldn't find related discussions.

**Strengths And Weaknesses:**

I like the idea of the paper and its results, while there are some unclear parts. I will raise my score once the authors resolve/answer the following issues/questions.
- Is there any reference/application of FGD (or FSGD)? If not, is there any theoretical implication of these results for GD (or SGD)?
- It seems that the proposed bounds can be vacuous due to the $\eta C_\eta$ multiplicative factor if the learning algorithms cannot achieve nearly zero training error. Is there optimal $\eta$ for the non-zero training error case which enables the bound non-vacuous?
- I am not sure if one can directly compare the GLD and SGLD bounds in the paper with those in [Negrea et al., 2019]. The authors applied Catoni's PAC-Bayesian bound to the result of [Negrea et al., 2019] and derived $O(\frac1{n-m}+\frac{(n-m)}{n^2}\sum_{t=1}^T\frac{\gamma_t^2}{\sigma_t^2}L^2)$. However, this bound and the bound in Theorem 6.2 may not be directly comparable since they bound different quantities; the first terms in RHS in Theorem 4.1 and Lemma 3.2 are different. I would like to see a more detailed discussion comparing the bounds in [Negrea et al., 2019] and this paper.
- Using the proposed GLD bound under mild assumptions including decaying step size (e.g., $\gamma_t\propto1/t$), Lipschitz continuity of $f$, etc., it seems that the generalization gap reduces to $O(1/(n-m))$, regardless of $T$. Using this reduction, can the excess risk bound also be derived?

---

> ### Author Response · Authors · 2022-08-02
> **Response to Reviewer Zg1r**
>
> Thanks for your valuable comments and enlightening questions! We address your questions as follows.
>
> 1. Is there any reference/application of FGD (or FSGD)? If not, is there any theoretical implication of these results for GD (or SGD)?
> Thanks for the question. Please see our response titled "Why study FGD/FSGD".
>
>
> 2. Question about $\eta C_{\eta}$: Indeed, $\eta C_{\eta}$ is at least $1$ when $\eta > 0$. Thus when the training error does not converge to 0, our test error bound does not converge to 0 either. But this does not imply that our theoretical bound will become vacuous ($\geq 1$). The best bound can be obtained by choosing suitable $\eta$. For instance, if one stops training when training error is $0.1$ and the term  $\frac{\ln(dT)}{n-m}\sum_{t=1}^T\left(\frac{\gamma_t^2}{\varepsilon_t^2}\left\|{g}_t\right\|^2\right)$ is approximately $0.1$. Then by choosing $\eta=1$, we still obtain a $0.317$ non-vacuous bound on the test error. The $\eta C_\eta$ appears in our theoretical results because we use the Catoni's form of Pac-Baysian bound. It enjoys the better $O(KL/n)$ rate compared with the classical $O(\sqrt{KL/n})$ Pac-Bayes bound. At a cost, it punishes the first empirical risk term $R(W,S)$ by a factor $\eta C_\eta>1$. However, when the training error is very small (e.g., many CV tasks), this cost is much smaller relative the other term. We remark that the $\eta$ cannot be optimized in advance as the training error $R(W,S_I)$ is unknown before training. However, there is a simple strategy on tuning $\eta$. Since $\eta C_\eta$ is an increasing function, we can choose small $\eta \rightarrow 0$ for the scenario where the training error is not so small.
>
> 3. Comparison with [Negrea et al., 2019]: Indeed, the comparison with their bound is not straightforward and requires a bit more derivation.
> Their work mainly studies in-expectation generalization bounds while we obtain high probability bounds with a constant factor associated with the training error, hence not directly comparable.
> For fair comparison, we can convert their in-expectation
> bound into a high-probability bound and use our version of Catoni's bound.
> We present the details in Appendix A in the revised version.
> Now, we briefly mention the details of the derivation.
> The idea is to apply the Theorem 4.1 in our paper (a data-dependent version of Catoni's bound). The sentence (``plug it into Catoni’s PAC-Bayesian bound'') in our original submission is indeed misleading and we have corrected it.
> First, Negrea et al. have shown in their paper that $KL(Q||P(S_J)) = O(\frac{(n-m)^2}{n^2}\sum_{t=1}^T\frac{L^2\gamma_t^2}{\sigma_t^2})$. Plugging it into Theorem 4.1, we have
> $R(W,D) \leq \eta C_{\eta} R(W,S_I) + C_{\eta}\frac{\ln(1/\delta)}{n-m} + C_{\eta}O(\frac{(n-m)}{n^2}\sum_{t=1}^T\frac{\gamma_t^2}{\sigma_t^2}L^2)$ holds w.p. $\geq 1-\delta$, which is an $O(\frac{1}{n-m} + \frac{n-m }{n^2}\sum_{t=1}^T\frac{\gamma_t^2}{\sigma_t^2}L^2)$ bound.
>
> 4. Let $\gamma_t$ be proportional to $1/t$: Thanks for the suggestion. Under this choice of learning rate and bounded gradient, the summation in our bound converges and can be bounded by a constant independent of $T$. Formally, we have the test error bound $R(W,D) \leq \eta C_\eta R(W,S_I) + O(\frac{1}{n-m})$. We have updated this part in Remark 6.4 of the revised version.
>
> 5. Limitations: Indeed, we did not explicitly mention the limitations of our results.
> The main limitation is that FGD requires computing the gradient difference for technical reasons. It would be more natural and desirable if we only need to compute the full gradient and rounded to the nearest grid point (or even better, to handle vanilla GD/SGD directly). However, there are subtle but challenging technical issues and we leave it as an important open problem. We have updated the conclusion section in the revised version.

---

> > ### Comment · Reviewer_Zg1r · 2022-08-06
> > **Response to Rebuttal**
> >
> > Thanks for your response. While I agree with other reviewers' concern that the connection between FGD and GD is weak, I still value the main idea presented in Theorem 4.1: splitting the training dataset for better bound. I raise my score to 5.

---

> > > ### Author Response · Authors · 2022-08-06
> > > **Response**
> > >
> > > Thanks a lot for carefully reading our response which we really appreciate.
> > > Indeed, FGD is still not GD, and whether one would think FGD is closely related to GD can be a subjective matter.
> > > Nevertheless, in our defense, we believe that FGD is close to GD from the following two perspectives
> > >
> > > (1) its definition  (just like 3.1415 is the finite precision version of $\pi$ ; in our case, we round the non-prior part of the gradient vector to certain precision)
> > >
> > > (2) its performance (its performace matches GD. See Experiment Figure 2(a),2(b) and Figure 3).
> > >
> > > More detailed technical motivation of FGD is provided in the separated response to all reviewers.
> > > In sum, we view our results for FGD/FSGD as novel results in this line of work and we chose to first present these results (before the results for stochastic Langevin dynamics that is more commonly studied).

---

### Official Review · Reviewer_nRQb · 2022-07-11

**Rating:** 5
**Confidence:** 3
**Soundness:** 3 good
**Presentation:** 2 fair
**Contribution:** 2 fair

**Summary:**

In this paper, the authors introduced a new discrete data-dependent prior to the PAC-Bayesian framework, and prove a high probability generalization based on Floored GD. The bound depends on the gradient difference $\|g_t\|$ between the full batch gradient and that of the prior samples. The empirical results showed that the presented bound could be non-vacuous.



**Questions:**

The presented bound seems to decrease first and then increase again (figure 1(b) and Figure 2(c)), even after the test error stabilizes after certain training steps. This would lead to a very loose and eventually vacuous bound, which is not ideal and becomes useless in the later training steps.  I am wondering how does the cumulative gradient difference change over the training epoch?


**Limitations:**

The experiments are very limited and do not have any comparison with the current available non-vacuous generalization bound.

**Strengths And Weaknesses:**

The FGD is used to derive the bound. While FGD has shown to have a similar learning curve as SGD (Figure 2 (a) vs. (b)), computing two gradients in a row with one of them to be a full batch gradient could be computationally expensive. While I understand the authors introduced FGD as an alternative to GD/SGD as a convenient way to analyze the bound, I do not think FGD/FSGD could become popular in practice and is not easy to understand intuitively.

The empirical evaluation is fairly weak. The bound on Cifar-10 looks fairly loose even though part of it is non-vacuous. While there exist bounds that are non-vacuous on Cifar-10 and are empirically shown to be much tighter than the presented bound, it is essential to provide a fair comparison to such bound to show the usefulness of the presented new bound.

There are a couple of typos in the paper:
Ln 85-86 "Floored Gradient Descent (FSGD), a variant of SGD." --> Floored Stochastic Gradient Descent (FSGD)
Figure 2 Caption * MNIST+SimpleNet+FSGD/SGD --> Should be CIFAR-10 not MNIST

---

> ### Author Response · Authors · 2022-08-02
> **Response to Reviewer nRQb**
>
> Thanks for your constructive comments, which lead to several improvements of our submission.
> We have fixed the typos and conducted several new experiments to address your concerns.
>
> Our responses to specific questions are as follows.
>
>
> 1. Why study FGD and FSGD? :
> Thanks for the question. Please see our response titled "Why study FGD/FSGD".
>
>
> 2. Experiment on CIFAR10: The reason why our bounds on CIFAR10 fall first and then rise is because we casually used a fixed learning rate rather than letting the learning rate decrease over time as in most of papers. If we multiply the learning rate by 0.9 every 200 steps, our test error bound always decreases and converges to 0.196 when the true test error is  0.18 (see Figure 2(c) in the revised version). This value is already smaller than the SOTA CIFAR10 bound of 0.23 reported in [1]. Thank you so much for inspiring us to improve our experiments!
>
> We also would like to explain why a decreasing learning rate is needed to avoid the first-decrease-then-increase phenomena. As in many trajectory-based analyses, the main term of the generalization bound is formed by the accumulation of the learning rate multiplying squared norm of a vector (e.g., gradient difference) at each step. During the training, the norm of the vector will not go to zero (see Figure 2(d) in our revised version and Figure 1 in [2]). Therefore, if the learning rate is fixed without decreasing, this type of upper bounds (including ours and similar bounds in previous paper for other algorithms such as SGLD) will increase by a constant at each step, resulting in the growth phenomenon in the original submission.
>
> 3. Tightness of our bound: We remark that our MNIST bound 0.037 is strictly tighter than the SOTA 0.11 reported in [1] (Figure 4). Moreover, after using decaying-learning rate schedule, our CIFAR10 bound 0.196 which is also smaller than the SOTA 0.23 reported in [1]. We also want to remark that our ultimate goal is to provide new theoretical techniques (constructing trajectory-based discrete prior) for analyzing discrete, non-random learning algorithms such as GD/SGD in non-convex setting. We believe our idea can inspire the future research along this direction.
>
> Reference
>
> [1] Dziugaite G K, Hsu K, Gharbieh W, et al. On the role of data in PAC-Bayes bounds[C]//International Conference on Artificial Intelligence and Statistics. PMLR, 2021: 604-612.
>
> [2] Haghifam, Mahdi, et al. "Sharpened generalization bounds based on conditional mutual information and an application to noisy, iterative algorithms." Advances in Neural Information Processing Systems 33 (2020): 9925-9935.

---

> > ### Comment · Reviewer_nRQb · 2022-08-08
> > **Thank you for your clarification**
> >
> > Thanks for your clarification on the connection between FGD and GD and additional efforts to improve experimental results. After reading your response, I still have concerns on FGD/FSGD:
> > 1. In line 100 to 101, you claim by "ignore the floor operation or let $\epsilon_t$ approaches 0, FGD reduces to GD". By looking at the Eq. (FGD), I feel it is not true, because GD is computed over $S$ not $S_J$. Correct me if I am wrong.
> > 2. For FSGD, since $g_2$ is computed over the intersection between $S_J$ and $B_t$ (Algorithm 2), how do you prevent it from getting empty set?
> > 3. The similarity between FGD and GD is done by eyeballing the training curve. I still feel this is very ad hoc.
> >
> > As a result, I am going to keep my score as it is.

---

> > > ### Author Response · Authors · 2022-08-09
> > > **Reply to "Thank you for your clarification"**
> > >
> > > Thanks for your response! We address your questions as follows.
> > >
> > > 1. FGD reduces to GD:
> > >
> > > This can actually be derived directly from the definition of FGD. Recall that the update rule of FGD is (WLOG let $\gamma_t=1$):
> > >
> > > $W_t \gets W_{t-1} - \nabla f(W_{t-1}, S_{J}) - \varepsilon_t\cdot \mathrm{floor}(\frac{\nabla f(W_{t-1}, S) - \nabla f(W_{t-1}, S_{J})}{\varepsilon_t})$.
> > >
> > > (1) If we ignore the floor operation in the last term, the equation becomes
> > > \begin{align*}
> > > W_t & \gets W_{t-1} - \nabla f(W_{t-1}, S_{J}) - \varepsilon_t\cdot (\frac{\nabla f(W_{t-1}, S) - \nabla f(W_{t-1}, S_{J})}{\varepsilon_t}) \\\\
> > > & =W_{t-1} - \nabla f(W_{t-1}, S_{J}) - (\nabla f(W_{t-1}, S) - \nabla f(W_{t-1}, S_{J})) \\\\
> > > & =W_{t-1} - \nabla f(W_{t-1}, S).
> > > \end{align*}
> > >
> > >  (2) When $\varepsilon \rightarrow 0$,
> > >  it is easy to see that
> > > $\lim_{\varepsilon \rightarrow 0} \varepsilon \cdot \mathrm{floor}(x/\varepsilon) = x$. Again, the FGD update rule reduces to
> > > \begin{align*}
> > > W_{t}  & \leftarrow W_{t-1} - \nabla f(W_{t-1}, S_{J}) - (\nabla f(W_{t-1}, S) - \nabla f(W_{t-1}, S_{J})) \\\\
> > > & =W_{t-1} - \nabla f(W_{t-1}, S).
> > > \end{align*}
> > >
> > > As we can see, the contribution of $\nabla f(W_{t-1}, S_J)$ has been canceled out. At this time, the FGD becomes GD. We felt the above derivation was too trivial to state explicitly in the paper.
> > >
> > > 2. "getting empty set": Yes. We remark that empty set is still valid in Algorithm 2. The gradient $\nabla f(w, \emptyset)$ is simply 0.
> > >
> > > 3. "The similarity between FGD and GD is done by eyeballing the training curve. I still feel this is very ad hoc."
> > >
> > > The training curves carry a lot of information such as the training loss and the convergence in practice. We could well draw a table or write some numbers down, but figures are more intuitive and informative (one can easily read off the numbers from the figures). Also the two training curves are very close. So close that we don't think we need another artificial measure to measure the closeness.
> > >
> > > Thanks for your reading. Please take our response into consideration.
> > > Should you have stronger reasons for the downright rejection rating, we are happy to answer them.
> > > (Reject: For instance, a paper with technical flaws, weak evaluation, inadequate reproducibility and incompletely addressed ethical considerations.) Our main contribution is theoretical, and we do not think our paper fits any of this.

---

### Author Response · Authors · 2022-08-02
**Why study FGD/FSGD?**

Some reviewers raised the following valid question: FGD/FSGD are not commonly used algorithms,
and they do not seem to perform any better than GD/SGD. So why bother study them and even prove generalization bounds for them?

We emphasize that we introduce and study FGD/FSGD, not because FGD/FSGD have better performance than GD/SGD or other advantages (they do not). Our reason is two-fold:
(1) FGD/FSGD are very close to and perform almost the same as GD/SGD (in terms of both optimization and generalization). So proving generalization error bounds for them
sheds light on the generalization performance of GD/SGD.
(2) More importantly, we introduce FGD/FSGD and use them as important stepping stones to extend existing mathematical technique for proving tighter generalization bounds for GD and SGD (which remains to be a major open problem in this line of research).

Now we elaborate on both points.
First, FGD and FSGD are finite precision variants of GD and SGD, obtained by rounding
part of the gradient to an integral multiply of the precision level $\varepsilon$.
As $\varepsilon$ approaches to 0, FGD/FSGD approaches to vanilla GD/SGD respectively.
Typically, $\varepsilon$ is chosen to be quite small and $\varepsilon=0.005$ in our experiments. Hence, the optimization and generalization performances of FGD and FSGD are almost the same as GD and SGD (verified by our experiments in Section 5 and Appendix G).
Since they are so close, we believe tight generalization error bounds for FGD/FSGD
are useful in understanding the generalization of GD/SGD.

Second, for technical reasons.
In recent years, an important line of work in generalization theory has focused
on proving tight generalization bounds for gradient methods such as GD/SGD/SGLD
for general non-convex loss functions,
via analyzing their trajectories (starting from Hardt et al.).
However, all existing proof techniques require
either restrictive assumptions on the learning rate (e.g., fast decreasing learning rate), or continuous injected noise (such as the Gaussian noise in Langevin dynamics)
for general non-convex loss functions. Hence, handling deterministic algorithms like GD or stochastic algorithms with discrete noises like SGD (without fast decreasing learning rate) is beyond existing techniques for proving generalization error bounds.
We introduce new technical ideas that can handle both FGD (a deterministic algorithm with updates very close to GD) and FSGD (a stochastic algorithm with discrete noise very similar to that of SGD), thus greatly extends the applicability of existing techniques.
In fact, understanding such discrete noises and their effect on generalization has been an important research topic (see e.g., [1][2][3]). In particular, [2] shows that it is not sufficient to approximate SGD's discrete noise by isotropic Gaussian noise. Moreover, proving nontrivial generalization bounds for SGD-like algorithms with discrete noise
has also been proposed as an open research direction in [1]. Although proving similar bounds for vanilla GD and SGD still requires overcoming subtle technical issue, we believe our new technique can be useful for proving tight generalization bounds for GD and SGD.

We add more discussions in both the introduction and the conclusion sections
in the revised version.

Reference

[1] Li, Jian, Xuanyuan Luo, and Mingda Qiao. "On Generalization Error Bounds of Noisy Gradient Methods for Non-Convex Learning." International Conference on Learning Representations. 2020.

[2] Zhu, Zhanxing, et al. "The Anisotropic Noise in Stochastic Gradient Descent: Its Behavior of Escaping from Sharp Minima and Regularization Effects." International Conference on Machine Learning. PMLR, 2019.

[3] Ziyin, Liu, et al. "Strength of Minibatch Noise in SGD." International Conference on Learning Representations. 2021.

---

### Meta-Review · Area_Chair_2u8z · 2022-08-26

**Recommendation:** Accept
**Confidence:** Less certain

**Metareview:**

Authors study generalization properties of gradient-based optimization algorithms via a PAC-Bayesian approach. Based on a data-dependent prior, authors establish a generalization bound for FGD, FSGD, GLD, and SGLD. The authors also provide convincing empirical studies to demonstrate that their results are not vacuous.

- Authors should better motivate the use of their seemingly synthetic algorithms.

- There are many typos in this paper, other than the ones listed by the reviewers. Although the reviewers did not raise this issue, authors should make sure their paper is ready for camera ready if the paper is accepted.

**Award:**

No

---

### Decision · Program_Chairs · 2022-09-14

Accept